# Likelihood Ratio Exponential Families

**Rob Brekelmans**[1], **Frank Nielsen** [2], **Alireza Makhzani** [3]
**Aram Galstyan**[1], **Greg Ver Steeg**[1]
[1]USC Information Sciences Institute, [2]Sony CSL, Tokyo
[3]University of Toronto, Vector Institute
{brekelma,galstyan,gregv}@isi.edu
makzhani@vectorinstitute.ai, frank.nielsen@acm.org

## Abstract

The exponential family is well known in machine learning and statistical physics as the maximum entropy distribution subject to a set of observed constraints [1], while the geometric mixture path is common in MCMC methods such as annealed importance sampling (AIS) [2, 3]. Linking these two ideas, recent work [4] has interpreted the geometric mixture path as an exponential family of distributions to analyse the thermodynamic variational objective (TVO) [5].

We extend *likelihood ratio exponential families* to include solutions to rate-distortion (RD) optimization [6, 7], the Information Bottleneck (IB) method [8], and recent rate-distortion-classification (RDC) approaches which combine RD and IB [9, 10]. This provides a common mathematical framework for understanding these methods via the conjugate duality of exponential families and hypothesis testing. Further, we collect existing results [11–14] to provide a variational representation of intermediate RD or TVO distributions as a minimizing an expectation of KL divergences. This solution also corresponds to a size-power tradeoff using the likelihood ratio test and the Neyman Pearson lemma. In thermodynamic integration (TI) bounds [15, 16] such as the TVO, we identify the intermediate distribution whose expected sufficient statistics match the log partition function.

## 1  Introduction

**Likelihood Ratio Exponential Family**    Following Grünwald [17] Ch. 19, or Brekelmans et al. [4], we consider the geometric mixture path between a base distribution $\pi_0(z)$ and a target $\pi_1(z)$, or posterior $\pi_1(z|x)$, as an exponential family. We define the sufficient statistics $\phi(z) = \log \pi_1(z)/\pi_0(z)$ as the log likelihood ratio [4], although in practice it is convenient to consider unnormalized distributions such as $\pi_1(z) \propto \tilde{\pi}_1(z)$ or $\pi_1(z|x) \propto \tilde{\pi}_1(x,z)$ and adjust the normalization constant accordingly. Using a natural parameter $\beta$ and base measure $\pi_0(z)$,

$$\pi_\beta(z) = \frac{1}{Z_\beta} \tilde{\pi}_0(z)^{1-\beta} \tilde{\pi}_1(z)^\beta \tag{1}$$

$$= \tilde{\pi}_0(z) \exp \left\{ \beta \cdot \phi(z) - \psi(\beta) \right\} \tag{2}$$

$$\text{where} \quad \phi(z) := \log \frac{\tilde{\pi}_1(z)}{\tilde{\pi}_0(z)} \qquad \psi(\beta) := \log Z_\beta = \log \int \tilde{\pi}_0(z)^{1-\beta} \tilde{\pi}_1(z)^\beta dz \,. \tag{3}$$

Before discussing examples in Sec. 2, we review background on conjugate duality in exponential families, which provides insights which are not evident from writing (2) as a geometric mixture [4].

Deep Learning through Information Geometry Workshop (NeurIPS 2020), Vancouver, Canada.

**Legendre Duality in Exponential Families** Since the log partition function $\psi(\beta)$ of an exponential family is a strictly convex, analytic function of the natural parameters $\beta$, its gradient will be unique and may be used as a dual parameterization for $\pi_\beta$ [18]. This diffeomorphism between the natural parameters $\beta = \{\beta_j\}$ [1] and moment parameters, denoted $\eta = \{\eta_j\}$, also defines the convex conjugate function $\psi^*(\eta)$, with

$$\psi^*(\eta) = \sup_\beta \beta \cdot \eta - \psi(\beta) \qquad \implies \qquad \eta_j = \frac{\partial \psi}{\partial \beta_j} = \mathbb{E}_{\pi_\beta}[\phi_j(z)] \ \forall \ j\,. \tag{4}$$

Using the Lebesgue or counting measure as $\pi_0(z)$, the conjugate $\psi^*(\eta)$ corresponds to the negative entropy of the maximum entropy solution $\pi_\beta(z)$ with observable constraint $\eta$ [18, 19]. With a general base measure (see App. A), we have

$$\psi^*(\eta_\beta) = D_{\mathrm{KL}}[\pi_\beta(z)||\pi_0(z)] \tag{5}$$

Since the convex conjugate is an involution $(\psi^*)^* = \psi$ by the Moreau biconjugation theorem [20], we can obtain a similar optimization for $\psi(\beta) = \sup_\eta \beta \cdot \eta - \psi^*(\eta)$. This leads to the canonical expression for Legendre duality, when the two optimizations are in equilibrium, and $\beta$ and $\eta_\beta$ are in correspondence (see App. B.1)

$$\psi(\beta) + \psi^*(\eta_\beta) - \beta \cdot \eta_\beta = 0\,. \tag{6}$$

Using any convex function, we can obtain a Bregman divergence via the first order Taylor remainder. For example, using $\psi(\beta)$ or $\psi^*(\eta)$, we have

$$D_\psi[\beta : \beta'] = \psi(\beta) - \psi(\beta') - \langle \beta - \beta', \nabla\psi(\beta')\rangle \qquad D_{\psi^*}[\eta : \eta'] = \psi^*(\eta) - \psi^*(\eta') - \langle \eta - \eta', \nabla\psi^*(\eta')\rangle\,.$$

With derivations in App. B, we can see that the Bregman divergences $D_\psi, D_{\psi^*}$ are equivalent with the order of the arguments reversed, and correspond to a KL divergence

$$D_\psi[\beta : \beta'] = D_{\mathrm{KL}}[\pi_{\beta'}||\pi_\beta] = D_{\psi^*}[\eta_{\beta'} : \eta_\beta]\,. \tag{7}$$

## 2 Examples

**Thermodynamic Variational Objective** In the variational autoencoder (VAE) setting, the TVO [5, 4] uses the approximate posterior as the initial distribution $\tilde\pi_0 = q(z|x)$ and joint generative model as the unnormalized target $\tilde\pi_1 = p_\theta(x, z)$.

$$\pi_\beta(z|x) = q(z|x) \exp\left\{ \beta \cdot \log \frac{p_\theta(x, z)}{q(z|x)} - \psi(x; \beta) \right\} \tag{8}$$

$$= \frac{1}{Z_\beta(x)} q(z|x)^{1-\beta} \, p_\theta(x|z)^\beta \tag{9}$$

with $\phi(z) = \log \tilde\pi_1/\tilde\pi_0$. Masrani et al. [5] use TI [15, 16] to express $\psi(x; 1) = \log Z_1(x) = \log p_\theta(x)$ as an integral over the geometric path,

$$\log Z_1(x) - \log Z_0(x) = \int_0^1 \frac{d}{d\beta} \log Z_\beta \, d\beta = \int_0^1 \mathbb{E}_{\pi_\beta}\big[\phi(x, z)\big] \, d\beta\,. \tag{10}$$

where we use the fact that the (partial) derivative of the log partition function equals the expected sufficient statistics in any exponential family [18]. Since $\psi(x; \beta)$ is convex in $\beta$ for any $x$, the left- and right-Riemann sums will provide lower and upper bounds on the log marginal likelihood,

$$\sum_{t=0}^{T-1} (\beta_{t+1} - \beta_t) \cdot \mathbb{E}_{\pi_{\beta_t}}\Big[\log \frac{p_\theta(x, z)}{q(z|x)}\Big] \leq \log Z_1 \leq \sum_{t=0}^{T-1} (\beta_{t+1} - \beta_t) \cdot \mathbb{E}_{\pi_{\beta_{t+1}}}\Big[\log \frac{p_\theta(x, z)}{q(z|x)}\Big]\,. \tag{11}$$

We derive novel insights on TVO curve via hypothesis testing in Sec. 3. Note that TI bounds as in (11) may be constructed for any one-dimensional likelihood ratio family with $\phi(z) = \log \tilde\pi_1/\tilde\pi_0$, such as in RD. However, more care would be required for multiple sufficient statistics as in RDC [9, 10].

---

[1] We allow for multiple sufficient statistics, with $\beta \cdot \phi(z) = \sum_j \beta_j \cdot \phi_j(z)$ denoting the dot product.

**Rate-Distortion**   Rate-distortion (RD) optimization ([6, 21, 8, 7] Ch. 13) formalizes the problem of lossy compression subject to a fidelity constraint. As in Alemi et al. [6][10], we measure the rate using the KL divergence to a fixed marginal distribution $\pi_0(z) = m(z)$, which upper bounds the mutual information in general. The distortion function $d(x, z)$ measures the quality of a code $z$. RD optimization seeks the minimum-rate encoding which achieves a desired average distortion $D$,

$$R(D) = \min_{q(z|x)} D_{\mathrm{KL}}[q(z|x)||m(z)] \quad \text{subj. to} \quad \mathbb{E}_{q(z|x)}[d(x|z)] \leq D \,. \tag{12}$$

We restrict our attention to a reconstruction loss distortion $d(x, z) = -\log p_\theta(x|z)$ as in [6]. Introducing $\beta$ to enforce the constraint, we obtain the unconstrained Lagrangian

$$\max_\beta \min_{q(z|x)} D_{\mathrm{KL}}[q(z|x)||m(z)] - \beta\big(\mathbb{E}_{q(z|x)}[d(x, z)] - D\big) \tag{13}$$

whose solution, for a given $m(z)$, has an exponential family form with $\phi(x, z) = -d(x, z)$ (e.g. [8])

$$\pi_\beta(z|x) = m(z)\,\exp\{-\beta \cdot d(x, z) - \psi(x; \beta)\} \tag{14}$$

$$= \frac{1}{Z_\beta(x)} m(z)\, p_\theta(x|z)^\beta \tag{15}$$

From the likelihood ratio perspective, we can choose $\pi_0(z) = m(z)$ and $\tilde{\pi}_1(x, z) = p_\theta(x|z)m(z) \propto p_\theta(z|x)$. Absorbing the factor of $p_\theta(x)$ into the normalizer $Z_\beta(x)$, we obtain the sufficient statistics

$$\phi(x, z) = \log\frac{\tilde{\pi}_1(x, z)}{\tilde{\pi}_0(z)} = \log\frac{p_\theta(x|z)m(z)}{m(z)} = \log p_\theta(x|z) = -d(x, z)\,, \tag{16}$$

so that the solution in (15) matches $\pi_\beta(z|x)$ in the likelihood ratio family induced by (16). The Lagrange multiplier $\beta$ is chosen to enforce the distortion constraint $D$, which, since $\phi(x, z) = -d(x, z)$, translates to seeking $\beta$ such that the moment parameters $\eta_\beta = -D$. At this optimal solution, $R(D)$ simply matches the conjugate $\psi^*(\eta)$ in (5)

$$R(D) = \psi^*(\eta) = D_{\mathrm{KL}}[\pi_\beta(z|x)||m(z)] \tag{17}$$

$$= \beta \cdot \eta - \psi(\beta) \tag{18}$$

$$= -\beta\, D - \log Z_\beta(x)\,. \tag{19}$$

Huang et al. [22] use the expression in (19) to estimate the RD curve using AIS [2].

**Information Bottleneck and RDC**   When defining 'relevant information' via a random variable such as a label $y$, the Information Bottleneck (IB) method [8, 23, 24] simplifies to an RD problem with a learned classifier providing the distortion function $c(y, z) = -\log p_\theta(y|z)$ ([8] or App.C).

$$\min_{q(z|x)} D_{\mathrm{KL}}[q(z|x)||m(z)] \quad \text{subj. to} \quad \mathbb{E}_{q(z|x)}[c(z, z)] \leq C \tag{20}$$

Recent work [9, 10] considers 'RDC' optimization using both reconstruction and classification loss,

$$\min_{q(z|x)} D_{\mathrm{KL}}[q(z|x)||m(z)] \quad \text{subj. to} \quad \mathbb{E}_{q(z|x)}[d(x, z)] \leq D \,, \ \mathbb{E}_{q(z|x)}[c(y, z)] \leq C \tag{21}$$

In this case, we may consider two sufficient statistics in our likelihood ratio exponential family. Similarly to multivariate IB [25, 26], we use an unnormalized target which factorizes as $\tilde{\pi}_1(x, y, z) = p_\theta(x|z)p_\theta(y|z)m(z)$, and consider the likelihood ratio sufficient statistics

$$\phi_d(x, z) = \log\frac{\pi_1(z|x)}{\pi_0(z)} = \log\frac{p_\theta(x|z)}{p_\theta(x)} \propto -d(x, z) \qquad \phi_c(y, z) = \log\frac{\pi_1(z|y)}{\pi_0(z)} = \log\frac{p_\theta(y|z)}{p(y)} \propto \log p_\theta(y|z) = -c(y, z)$$

where we again absorb $p_\theta(x)$ and $p(y)$ into the normalization. Introducing Lagrange multipliers $\beta = \{\beta_D, \beta_C\}$ to enforce $\eta_d(\beta) = -D$, $\eta_c(\beta) = -C$ at optimality, we obtain the solution of (21) as a geometric mixture [9, 10] belonging to the likelihood ratio family with $\phi = \{\phi_d, \phi_c\}$

$$\pi_\beta(z|x, y) = m(z)\exp\big\{\beta_D \cdot \phi_d(x, z) + \beta_C \cdot \phi_c(y, z) - \psi(x, y; \beta)\big\} \tag{22}$$

$$= \frac{1}{Z_\beta(x, y)} m(z)\, p_\theta(x|z)^{\beta_D}\, p_\theta(y|z)^{\beta_C}$$

With applications in transfer learning, Gao and Chaudhari [9] seek to evolve the model parameters $\theta$ and approximate posterior $q(z|x)$ along the 'equilibrium surface' of optimal solutions to (21). We

interpret their free energy $F(\beta_D, \beta_C)$ as the negative log partition function $-\psi(\beta_D, \beta_C)$, where $\beta_D, \beta_C$ are analogous to the *intensive* variables of a physical system [10]. Written using the conjugate optimization (4), we seek $\{\theta, q(z|x)\}$ that yield the appropriate distortion and classification loss

$$-F(\beta_D, \beta_C) = \psi(\beta_D, \beta_C) = \sup_{\eta_d, \eta_c} \beta_D\, \eta_d + \beta_C\, \eta_c - \psi^*(\eta_d, \eta_c) \tag{23}$$

Similarly, for given *extensive* variables $\eta_D, \eta_C$, the optimal rate $R(D, C)$ corresponds to $\psi^*(\eta_D, \eta_C)$

$$R(D, C) = \psi^*(\eta_D, \eta_C) = \sup_{\beta_d, \beta_c} -\beta_d\, D - \beta_c\, C - \psi(\beta_d, \beta_c)\,, \tag{24}$$

At optimality on the 'equilibrium surface' [9], we obtain equality in the expression for Legendre duality (6). In other words, for the current decoder and classifier parameters $\theta$, the encoder $q(z|x)$ matches $\pi_\beta(z|x)$ in the likelihood ratio family (22), with $\beta = \{\beta_D, \beta_C\}$. This distribution fulfills the constraints $\eta_\beta = \{\eta_D, \eta_C\} = \{-D, -C\}$, so that

$$\psi(\beta_D, \beta_C) + \psi^*(\eta_D, \eta_C) - \beta_D\, \eta_D - \beta_C\, \eta_C = 0\,. \tag{25}$$

This expression (25) also translates to the 'first law of learning' from Alemi and Fischer [10], when $\psi(\beta_D, \beta_C)$ is considered a fixed quantity for given a choice of $\beta_D, \beta_C$.

## 3  Variational Representations and Hypothesis Testing

Grosse et al. [12] note that any distribution along the geometric mixture path can be given a variational representation as the solution to an expected KL divergence minimization

$$\pi_\beta(z) = \arg\min_{r(z)} (1 - \beta)\, D_{\mathrm{KL}}[r(z)||\pi_0(z)] + \beta\, D_{\mathrm{KL}}[r(z)||\pi_1(z)] \tag{26}$$

We proceed to interpret (26) as a Bregman information (or gap in Jensen's inequality) [11], or as describing an optimal decision rule for hypothesis testing using the Neyman Pearson lemma. We restrict our attention to a one-dimensional likelihood ratio family, as in TVO, RD, or IB, in this section.

**Bregman Information**  Banerjee et al. [11] define the *Bregman information* as the minimum expected Bregman divergence to a representative point in the second argument. Regardless of the Bregman generator, the optimal representative corresponds to the mean over the input arguments. Since $D_{\mathrm{KL}}[r_\beta(z)||\pi_0(z)] = D_\psi[0 : \beta_r]$ when optimizing over $r_\beta(z)$ in the exponential family, we can rewrite (26) as

$$\beta = \arg\min_{\beta_r} (1 - \beta)\, D_\psi[0 : \beta_r] + \beta\, D_\psi[1 : \beta_r] \; = \; (1 - \beta) \cdot 0 + \beta \cdot 1 \tag{27}$$

At this optimum, the expected KL divergence (27) can be written as a gap in Jensen's inequality for the convex function $\psi(\beta)$ [11],

$$\mathcal{J}_{\psi, \{1-\beta, \beta\}, \{0,1\}} = (1 - \beta)\, D_\psi[0 : \beta] + t\, D_\psi[1 : \beta] \tag{28}$$

$$= (1 - \beta)\, \psi(0) + \beta\, \psi(1) - \psi(\beta) \tag{29}$$

We visualize this gap in Jensen's inequality in Fig. 2. As shown in [27] or App. E.1, we can also view $\mathcal{J}_\psi$, or the expected KL divergence (26), as a Rényi divergence with order $\beta$

$$\mathcal{J}_{\psi, \{1-\beta, \beta\}, \{0,1\}} = (1 - \beta)\, D_{\mathrm{KL}}[\pi_\beta(z)||\pi_0(z)] + \beta\, D_{\mathrm{KL}}[\pi_\beta(z)||\pi_1(z)] \tag{30}$$

$$= (1 - \beta)\, D_\beta[\pi_1(z) : \pi_0(z)]\,.$$

Grünwald [17] and Harremoës [28] provide additional coding interpretations of the Rényi divergence.

**Neyman Pearson Lemma**  Suppose we have access to $n$ i.i.d. observations from an unknown distribution $r(z)$, and are interested in testing the hypotheses that either $H_0 : r(z) = \pi_0(z)$ or $H_1 : r(z) = \pi_1(z)$. The Neyman-Pearson lemma states that the likelihood ratio test is optimal, in the sense that, for any other decision region with type-1 error $Pr(e_1) = R$, then the type-2 error is no better than that of the likelihood ratio test ([7] Ch. 11, [14])[2]. The decision rule is given by

$$A_n(\pi_1; \eta) = \left\{ z_{1:n} \;\middle|\; \frac{1}{n} \sum_{i=1}^{n} \log \frac{\pi_1(z_i)}{\pi_0(z_i)} \geq \eta \right\} \tag{31}$$

---

[2]While Neyman-Pearson is often obtained via the method of types [7], Csiszár [29] treat the continuous case.

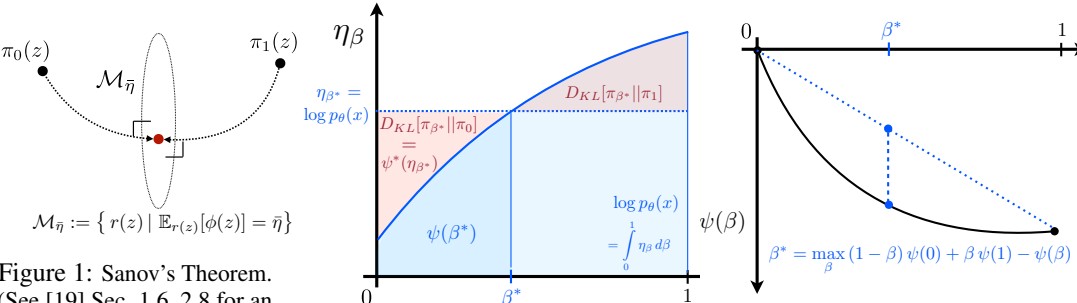

Figure 1: Sanov's Theorem. (See [19] Sec. 1.6, 2.8 for an interpretation in terms of projection and a generalization of the Pythagorean Theorem)

Figure 2: Chernoff point on $\eta_\beta = \nabla\psi(\beta)$.

Figure 3: Chernoff point on $\psi(\beta)$

for some threshold $\eta$. Let a type-1 error occur when $n$ i.i.d. draws $\{z_i\}_{i=1}^N$ from $\pi_0(z)$ will yield empirical expectations exceeding the threshold $\eta$. Sanov's Theorem and large deviation theory ([7] Ch. 11, [30]) states that the asymptotic error exponent corresponds to a KL divergence

$$\lim_{n\to\infty} \frac{1}{n}\Pr(e_1) \to \exp\{-D_{\mathrm{KL}}[r^*(z)||\pi_0(z)]\}, \tag{32}$$

$$\text{where } r^*(z) = \min_{r(z)\in\mathcal{M}_\eta} D_{\mathrm{KL}}[r(z)||\pi_0(z)] \tag{33}$$

The feasible set $\mathcal{M}_\eta := \{r(z) \mid \mathbb{E}_r \log \frac{\pi_1(z)}{\pi_0(z)} = \eta\}$ reflects a expectation constraint corresponding to a given decision threshold, and the error exponent is obtained by minimizing the divergence subject to this constraint. With $\psi^*(\eta) = D_{\mathrm{KL}}[\pi_{\beta_\eta}(z)||\pi_0(z)]$ as in (5), this exactly matches the conjugate or maximum entropy optimization for a given expected sufficient statistic $\eta$, and thus $r^*(z)$ lies within the likelihood ratio exponential family,

$$r^*(z) = \pi_0(z)\exp\left\{\beta_\eta \cdot \log\frac{\pi_1(z)}{\pi_0(z)} - \psi(\beta)\right\} \tag{34}$$

As shown in Fig. 1, Sanov's Theorem implies a similar expression for the asymptotic type-2 error, when draws from $\pi_1(z)$ achieve a *lower* expected likelihood ratio than $\eta$. Expressing the conditions of the Neyman Pearson lemma using these asymptotic error probabilities, we can write

$$\Pr(e_2) = \min_{r(z)} D_{\mathrm{KL}}[r(z)||\pi_1(z)] \quad \text{subj. to} \quad D_{\mathrm{KL}}[r(z)||\pi_0(z)] = R \tag{35}$$

Using a Lagrange multiplier $\lambda = \frac{1-\beta}{\beta}$ to enforce the constraint, we obtain the variational form (26)

$$\frac{1}{\beta}\Pr(e_2) = \min_{r(z)}(1-\beta)D_{\mathrm{KL}}[r(z)||\pi_0(z)] + \beta D_{\mathrm{KL}}[r(z)||\pi_1(z)] \tag{36}$$

Thus, any distribution in our likelihood ratio exponential family corresponds to a likelihood ratio test with decision threshold $\eta$, which is optimal for a type-1 error region of size $\psi^*(\eta) = R$.

**Chernoff Information** While each choice of $\beta_\eta$ determines a likelihood ratio test and error region, how should we choose this parameter? Regardless of the prior probabilities $p_0, p_1$ that we might assign to each hypothesis in a Bayesian setting, the Chernoff information provides the best achievable error exponent in the large sample limit ([13], [7] Ch. 11).

$$C^* = -\min_\beta \log \int \pi_0(z)^{1-\beta}\pi_1(z)^\beta dz \tag{37}$$

$$= -\max_\beta (1-\beta)\psi(0) + \beta\psi(1) - \psi(\beta) \tag{38}$$

To rewrite (37) as a gap in Jensen's inequality (38), we have expanded $\pi_0 = \tilde\pi_0/Z_0$ and $\pi_1 = \tilde\pi_1/Z_1$, with $\psi(\beta) = \log \int \tilde\pi_0(z)^{1-\beta}\tilde\pi_1(z)^\beta dz$.

We denote the optimum over $\beta$ as the *Chernoff point* [13], or $\beta^*$. In App. E.2, we derive the moment-matching condition

$$\eta_{\beta^*} = \frac{\psi(\beta_1) - \psi(\beta_0)}{\beta_1 - \beta_0} \qquad (39)$$

which holds between arbitrary $\beta_0, \beta_1$ and implies $\eta_{\beta^*} = \psi(1) - \psi(0)$ for $\beta_0 = 0, \beta_1 = 1$. At this critical point, we show in App. E.3 that the KL divergence to the endpoints is equal,

$$D_{\mathrm{KL}}[\pi_{\beta^*}(z)||\pi_0(z)] = D_{\mathrm{KL}}[\pi_{\beta^*}(z)||\pi_1(z)]. \qquad (40)$$

**Chernoff Point on the TVO Integrand**   For the unnormalized likelihood ratio $\log \tilde\pi_1(z)/\pi_0(z)$, we can interpret the Chernoff point using thermodynamic integration bounds (11)

$$\sum_{t=0}^{T-1} (\beta_{t+1} - \beta_t) \cdot \mathbb{E}_{\pi_{\beta_t}} \Big[ \log \frac{\tilde\pi_1(x,z)}{\tilde\pi_0(z)} \Big] \leq \log Z_1 \leq \sum_{t=0}^{T-1} (\beta_{t+1} - \beta_t) \cdot \mathbb{E}_{\pi_{\beta_{t+1}}} \Big[ \log \frac{\tilde\pi_1(x,z)}{\tilde\pi_0(z)} \Big] \qquad (41)$$

With $\pi_0(z) = q(z|x)$ as in TVO [5, 4], we note that the integrand at $\beta_0 = 0$ corresponds to the familiar evidence lower bound (ELBO), $\mathbb{E}_{\pi_0} \big[ \log \frac{\tilde\pi_1(x,z)}{\pi_0(z|x)} \big] = \log Z_1(x) - D_{\mathrm{KL}}[\pi_0(z|x)||\pi_1(z|x)]$. Similarly, at $\beta_1 = 1$, the integrand $\mathbb{E}_{\pi_1}[\cdot] = \log Z_1(x) + D_{\mathrm{KL}}[\pi_1(z|x)||\pi_0(z|x)]$ is an upper bound. Since $\psi(1) = \log p_\theta(x)$ and $\psi(0) = 0$, the condition for the Chernoff point in (39) corresponds to

$$\eta_{\beta^*} = \mathbb{E}_{\pi_{\beta^*}} \Big[ \log \frac{p_\theta(x,z)}{q(z|x)} \Big] = \log p_\theta(x), \qquad (42)$$

or the point after which the expected likelihood ratio switches from an lower bound to an upper bound. We visualize this in Fig. 2, with $\eta_{\beta^*}$, as a point on the y-axis, equal to the area under the curve, $\log p_\theta(x)$. Note that the red shaded regions correspond to the KL divergence from $\pi_{\beta^*}$ to each endpoint (see [4]), and will have equal area due to (40).

# 4   Conclusion

We have presented likelihood ratio exponential families as a common framework for understanding TVO, RD, IB, and RDC optimizations in terms of conjugate duality and hypothesis testing. These insights may be useful for improving mutual information estimators which leverage intermediate distributions [31], learn a binary classifier distinguishing samples from $\tilde\pi_0, \tilde\pi_1$ [32, 33], or involve a neural network 'critic' whose optimal function output is the true likelihood ratio [34].

While it is natural to introduce additional sufficient statistics from the exponential family perspective, thermodynamic integration bounds and hypothesis testing interpretations remain to be clarified in higher dimensions as in RDC. Further exploring the approach of Gao and Chaudhari [9], for evolving model parameters and Lagrange multipliers along the equilibrium surface of solutions to Eq. (6), is an exciting future direction. Beyond the applications shown in [9], this could lead to replacing heuristics such as KL annealing in $\beta$-VAE with more principled dynamics for $\beta$ over the course of optimization.

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

## A Conjugate as a KL Divergence

When considering an exponential family of the form

$$\pi_\beta(z) = \pi_0(z) \exp\{\beta \cdot \phi(z) - \psi(\beta)\}. \tag{43}$$

we show that $\psi^*(\eta)$ takes the form of a KL divergence when considering a base measure $\pi_0(z)$.

$$\psi^*(\eta) = \sup_\beta \beta \cdot \eta - \psi(\beta) \tag{44}$$

$$= \beta_\eta \cdot \eta - \psi(\beta_\eta)$$

$$= \mathbb{E}_{\pi_{\beta_\eta}}[\beta_\eta \cdot \phi(z)] - \psi(\beta_\eta)$$

$$= \mathbb{E}_{\pi_{\beta_\eta}}[\beta_\eta \cdot \phi(z)] - \psi(\beta_\eta) \pm \mathbb{E}_{\pi_{\beta_\eta}}[\log \pi_0(z)]$$

$$= \mathbb{E}_{\pi_{\beta_\eta}}[\log \pi_{\beta_\eta(z)} - \log \pi_0(z)]$$

$$= D_{KL}[\pi_{\beta_\eta}(z) \| \pi_0(z)] \tag{45}$$

where we have added and subtracted a factor of $\mathbb{E}_{\pi_{\beta_\eta}} \log \pi_0(z)$ in the fourth line. If $\pi_0(z)$ is constant with respect to $z$ using, for example, the uniform measure, then $D_{KL}[\pi_{\beta_\eta}(z) \| \pi_0(z)]$ reduces to the familiar definition of the conjugate $\psi^*(\eta)$ as the negative entropy $\mathbb{E}_{\pi_{\beta_\eta}} \log \pi_{\beta_\eta}(z)$ [18].

## B KL Divergence as a Bregman Divergence

For an exponential family with partition function $\psi(\beta)$ and sufficient statistics $\phi(z)$ over a random variable $z$, the Bregman divergence $D_\psi$ corresponds to a KL divergence. Recalling that $\nabla_\beta \psi(\beta) = \eta_\beta = \mathbb{E}[\phi(z)]$, we simplify the definition of the Bregman divergence to obtain

$$D_\psi[\beta : \beta'] = \psi(\beta) - \psi(\beta') - \beta \cdot \eta_{\beta'} + \beta' \cdot \eta_{\beta'}$$

$$= \psi(\beta) - \psi(\beta') - \mathbb{E}_{\pi_{\beta'}}[\beta \cdot \phi(z)] + \mathbb{E}_{\pi_{\beta'}}[\beta' \cdot \phi(z)]$$

$$= \underbrace{\mathbb{E}_{\pi_{\beta'}}[\beta' \cdot \phi(z) - \psi(\beta')] + \mathbb{E}_q[\pi_0(z)]}_{\log \pi_{\beta'}(z)} - \underbrace{\mathbb{E}_{\pi_{\beta'}}[\beta \cdot \phi(z) - \psi(\beta)] - \mathbb{E}_q[\pi_0(z)]}_{\log \pi_\beta(z)}$$

$$= \mathbb{E}_{\pi_{\beta'}}[\log \pi_{\beta'}(z) - \log \pi_\beta(z)]$$

$$= D_{KL}[\pi_{\beta'}(z) \| \pi_\beta(z)] \tag{46}$$

where we have added and subtracted terms involving the base measure $\pi_0(z)$, and used the definition of our exponential family from (43). The Bregman divergence $D_\psi$ is thus equal to the KL divergence with arguments reversed.

### B.1 Canonical Divergence

We can also show that the Bregman divergences $D_\psi, D_{\psi^*}$ are equivalent up to reordering of the arguments

$$D_\psi[\beta : \beta'] = D_{\psi^*}[\eta' : \eta] \tag{47}$$

where we abbreviate $\eta' = \eta_{\beta'}$ and $\eta = \eta_\beta$. The conjugacy relationships

$$\psi^*(\eta') = \beta' \cdot \eta' - \psi(\beta') \qquad \psi(\beta) = \beta \cdot \eta - \psi^*(\eta) \tag{48}$$

can be used to translate between these dual divergences.

$$D_\psi[\beta : \beta'] = \psi(\beta) - \psi(\beta') - \beta \cdot \eta' + \beta' \cdot \eta'$$

$$= \psi(\beta) + \psi^*(\eta') - \beta \cdot \eta' \tag{49}$$

$$= \psi^*(\eta') - \psi^*(\eta) + \beta \cdot \eta - \beta \cdot \eta'$$

$$= \psi^*(\eta') - \psi^*(\eta) - \langle \eta' - \eta, \nabla \psi^*(\eta) \rangle$$

$$= D_{\psi^*}[\eta' : \eta]$$

The intermediate expression (49) is known as the canonical form of the divergence [19]

$$\psi(\beta) + \psi^*(\eta') - \beta \cdot \eta' = D_\psi[\beta : \beta'] = D_{\psi^*}[\eta' : \eta] \tag{50}$$

Comparing with the expression for Legendre duality in (6), note that the correspondence between $\beta$ and $\eta$ implies that the divergence vanishes, since both parameterizations refer to the same distribution

$$\psi(\beta_\eta) + \psi^*(\eta_\beta) - \beta_\eta \cdot \eta_\beta = 0 \implies D_\psi[\beta_\eta : \beta_\eta] = D_{\psi^*}[\eta_\beta : \eta_\beta] = 0 \tag{51}$$

## C  Information Bottleneck as Rate-Distortion

The Information Bottleneck (IB) method [8] defines the 'relevant information' in a representation, $I(Y : Z)$, via another variable of interest $Y$, often taken to be a label. The IB objective then seeks a minimal encoding $Z$ which maintains a given level of predictive ability about the target.

$$\min_{q(z|x)} I_q(X; Z) \text{ subj. to. } I_q(Y; Z) \geq I_c \tag{52}$$

where we let $I_q$ reflect the exact mutual information for the true data and label distributions $q(x)q(y|x)$ with a given encoding function $q(z|x)$.

When the desired information constraint equals the total information $I_c = I_q(X; Y)$ that the data source contains about the label, (52) corresponds to the problem of finding the minimal sufficient statistics $z$ for $y$ with respect to $x$. The IB objective generalizes this optimization for smaller values of $I_c$.

Since $I_q(Y; Z) = H_q(Y) - H_q(Y|Z) = -\mathbb{E}_q \log q(y) + \mathbb{E}_q \log q(y|z)$, we can ignore the label entropy as a constant with respect to $z$. While it may be difficult to obtain the true posterior $q(y|z)$ of the labels given latent variables , we can instead optimize a variational classifier $p(y|z)$. This provides an lower bound on the mutual information since $D_{KL}[q(y|z)||p(y|z)] \geq 0$ and is also known as the 'test channel' in rate-distortion theory ([7] Ch. 13). Applying this inequality within the unconstrained IB Lagrangian,

$$\mathcal{L}_{IB} = \max_\beta \min_{q(z|x)} I_q(X; Z) - \beta \left( -\mathbb{E}_q \log q(y) + \mathbb{E}_q \log q(y|z) - I_c \right)$$

$$\geq \max_\beta \min_{q(z|x), p(y|z)} I_q(X; Z) - \beta \left( -\mathbb{E}_q \log q(y) + \mathbb{E}_q \log p(y|z) - I_c \right)$$

$$= \max_\beta \min_{q(z|x), p(y|z)} I_q(X; Z) - \beta \, \mathbb{E}_{p(y(x), z)}[p(y|z)] + \text{const} \tag{53}$$

where $y(x)$ indicates the label of a given data point.

As shown in Tishby et al. [8], the Information Bottleneck is a special case of rate-distortion with

$$c(y(x), z) = D_{KL}[q(y|x)||q(y|z)] = \mathbb{E}_q[q(y|x)] - \mathbb{E}_q[q(y|z)] \tag{54}$$

Comparing (53) with (54), note that $\mathbb{E}_q[q(y|x)]$ is a constant, leaving $c(y(x), z) = -\mathbb{E}_{q(y(x)|z)}[q(y|z)]$ as the effective distortion measure. If this quantity is intractable, we can instead define the distortion function using $p(y|z)$ as above.

## D  Bregman Information and Jensen Gaps

Imagine we are interested in minimizing the expected Bregman divergence $D_f$ to a single representative point, which may then be thought of as the optimal codeword for $R = 0$ in a rate-distortion scenario using a Bregman divergence distortion. Banerjee et al. [11] show that, regardless of the divergence, the minimizing point will be the mean with respect to a desired measure, and the expected divergence will be a gap in Jensen's inequality for the function $f$.

**Theorem D.1** (Bregman Information, [11]). *Let $X$ be a random variable that takes values in $\mathcal{X} = \{x_i\}_{i=1}^n \subseteq \mathcal{S} \subseteq \mathbb{R}^d$ following a positive probability measure $\nu$ such that $\mathbb{E}_\nu[X] \in ri(\mathcal{S})$. Given a Bregman divergence $D_f : \mathcal{S} \times ri(\mathcal{S}) \mapsto [0, \infty]$, the problem:*

$$\mathcal{J}_{\nu,f}(s^*) = \min_{s \in ri(\mathcal{S})} \mathbb{E}_\nu[D_f(X, s)]$$

*has a unique minimizer given by the mean $s^* = \mu = \mathbb{E}_\nu[X]$. At this $\arg\min$, $\mathcal{J}_{\nu,f}(s^*)$ corresponds to a gap in Jensen's inequality for the convex function $f$ and expectations with respect to $\nu$.*

$$\mathbb{E}_\nu[f(X)] - f(\mathbb{E}_\nu[X]) = \mathcal{J}_{\nu,f}(s^*)$$

*Proof.* Consider a point $s$ and the mean $\mu = \mathbb{E}_\nu[X]$, both in $ri(\mathcal{S})$ so that $\mathcal{J}_{\nu,f}$ is well defined.

$$
\begin{aligned}
\mathcal{J}_{\nu,f}(s) - \mathcal{J}_{\nu,f}(\mu) &= \mathbb{E}_\nu D_f(x,s) - \mathbb{E}_\nu D_f(x,\mu) \\
&= \cancel{\mathbb{E}_\nu f(x)} - f(s) - \langle \mathbb{E}_\nu x - s, \nabla f(s) \rangle \\
&\quad - \cancel{\mathbb{E}_\nu f(x)} + f(\mu) - \langle \mathbb{E}_\nu x - \mu, \nabla f(\mu) \rangle \\
&= f(\mu) - f(s) - \langle \mu - s, \nabla f(s) \rangle \\
&= D_f[\mu, s] \geq 0
\end{aligned}
$$

with equality only when $s = \mu = \mathbb{E}_\nu[X]$ if $f$ is strictly convex. Then,

$$
\begin{aligned}
\mathcal{J}_{\nu,f}(\mu) &= \mathbb{E}_\nu[D_f(X,\mu)] \\
&= \mathbb{E}_\nu[f(X)] - f(\mu) - \underbrace{\mathbb{E}_\nu \langle x - \mu, \nabla_x f(\mu) \rangle}_{0} \\
&= \mathbb{E}_\nu[f(X)] - f(\mu)
\end{aligned}
$$

which amounts to a gap in Jensen's inequality for the function $f$ and measure $\nu$. $\qquad\square$

## E   Gap in Jensen's Inequality for $\psi(\beta)$

In this section, we analyse the Bregman Information and Jensen gap associated with the log partition function of the likelihood ratio exponential family. As in Sec. 3, this corresponds to the variational representation of Grosse et al. [12], while maximizing the Jensen gap will lead to the Chernoff point. We give proofs of intermediate results at the end of the section.

With $D_\psi$ as the divergence associated with the convex function $f(X) = \psi(\beta)$, we take the expected Bregman divergence using a convex combination ($\{1 - \alpha, \alpha\}$) over arguments $\{\beta_0, \beta_1\}$.

$$
\mathcal{J}_{\psi,\alpha} = \min_{\beta'} (1 - \alpha) D_\psi[\beta_0 : \beta'] + \alpha D_\psi[\beta_1 : \beta'] \tag{55}
$$

$$
\mu_\beta^{(\alpha)} = (1 - \alpha)\, \beta_0 + \alpha\, \beta_1 \tag{56}
$$

where Theorem D.1 shows that the minimizer $\mu_\beta^{(\alpha)}$ occurs at the mean of the arguments. After simplifying, we can see that this corresponds to a Jensen's inequality for the convex function $\psi(\beta)$

$$
\mathcal{J}_{\psi,\alpha} = (1 - \alpha)\, D_\psi[\beta_0 : \mu_\beta^{(\alpha)}] + \alpha\, D_\psi[\beta_1 : \mu_\beta^{(\alpha)}] \tag{57}
$$

$$
= (1 - \alpha)\, \psi(\beta_0) + \alpha\, \psi(\beta_1) - \psi\big((1 - \alpha)\beta_0 + \alpha\, \beta_1\big) \tag{58}
$$

For the case of $\beta_0 = 0$ and $\beta_1 = 1$, we see that the optimal parameter is simply $\mu_\beta^{(\alpha)} = \alpha$ so that

$$
\mathcal{J}_{\psi,\alpha} = (1 - \alpha)\, \psi(0) + \alpha\, \psi(1) - \psi(\alpha) \tag{59}
$$

Nielsen and Nock [27] demonstrate the following lemma, showing the relationship between the Jensen's gap and Rényi divergence within an exponential family.

**Lemma E.1.** *The Rényi divergence of order $\alpha$ between two distributions (indexed by natural parameters $\beta_0$ and $\beta_1$) within an exponential family (with log partition function $\psi(\beta)$) has the form of a gap in Jensen's inequality $\mathcal{J}_{\psi,\{1-\alpha,\alpha\},\{\beta_0,\beta_1\}}$, abbreviated $\mathcal{J}_{\psi,\alpha}$*

$$
(1 - \alpha) D_\alpha[\pi_{\beta_0}(z) : \pi_{\beta_1}(z)] = -\log \int \pi_{\beta_0}(z)^{1-\alpha} \pi_{\beta_1}(z)^\alpha dz \tag{60}
$$

$$
= (1 - \alpha)\, \psi(\beta_0) + \alpha\, \psi(\beta_1) - \psi\big((1 - \alpha)\beta_0 + \alpha\, \beta_1\big) \tag{61}
$$

$$
= \mathcal{J}_{\psi,\alpha} \tag{62}
$$

*Proof.* The first equality follows from the definition of $D_\alpha$, and corresponds to the Chernoff coefficient in (37) or Jensen gap $\mathcal{J}_{\psi,\alpha}$. We demonstrate the second equality for the likelihood ratio family in App. E.1, or see Nielsen and Nock [27] for the general case. $\qquad\square$

Van Erven and Harremos [35] show that the scaled Renyi divergence $(1-\alpha)D_\alpha[\pi_{\beta_0} : \pi_{\beta_1}]$ is concave. For given endpoint distributions, we can thus seek to maximize $\mathcal{J}_{\psi,\alpha}$ as a function of $\alpha$.

**Lemma E.2.** *Maximizing the Jensen's gap $\mathcal{J}_{\psi,\alpha}$ obtained from arguments $\{\beta_0, \beta_1\}$ of the convex function $\psi(\beta)$, with respect to the choice of mixing weight $\{1 - \alpha, \alpha\}$,*

$$\arg\max_{\alpha} \mathcal{J}_{\psi,\alpha} = \arg\max_{\alpha}(1 - \alpha)\,\psi(\beta_0) + \alpha\,\psi(\beta_1) - \psi\big((1 - \alpha)\beta_0 + \alpha\,\beta_1\big) \tag{63}$$

*leads to the following condition*

$$\eta_{\alpha^*} = \frac{\psi(\beta_1) - \psi(\beta_0)}{\beta_1 - \beta_0} \tag{64}$$

*Proof.* See App. E.2 for proof. $\qquad\square$

For $\beta_0 = 0$ and $\beta_1 = 1$, this suggests that the expected sufficient statistics $\eta_\alpha$ (with natural parameter $\mu_\beta^{(\alpha)} = \alpha$) should match the marginal likelihood $\log p_\theta(x)$.

$$\eta_\alpha = \frac{\psi(1) - \psi(0)}{1 - 0} = \log p_\theta(x) \tag{65}$$

**Lemma E.3.** *At the maximum in Eq. (63), consider the distribution $\pi_{\beta_\alpha^*}$ in the same exponential family, with natural parameter $\beta_\alpha^* = \mu_\beta^{(\alpha)} = (1 - \alpha)\,\beta_0 + \alpha\,\beta_1$, the Bregman divergences to each endpoint $\{\beta_0, \beta_1\}$ are the same.*

$$D_\psi[\beta_0 : \beta_{\alpha^*}] = D_\psi[\beta_1 : \beta_{\alpha^*}] \tag{66}$$

$$\tag{67}$$

*Since the Bregman divergence $D_\psi$ within an exponential family corresponds to the KL divergence, we can equivalently write*

$$D_{KL}[\pi_{\beta_{\alpha^*}} : \pi_{\beta_0}] = D_{KL}[\pi_{\beta_{\alpha^*}} : \pi_{\beta_1}] \tag{68}$$

*Proof.* See App. E.3 for Bregman divergence derivations. See discussion around (7) for the relationship between the Bregman and KL divergence. $\qquad\square$

**Dual Jensen Gap using $\psi^*(\eta)$**   Note that we could also construct a Jensen gap from the dual divergence $\psi^*(\eta_\beta) = D_{KL}[\pi_\beta||\pi_0]$, with $D_{\psi^*}[\eta : \eta'] = D_{KL}[\pi_\eta||\pi_{\eta'}]$

$$\mathcal{J}_{\psi^*,\lambda} = \lambda\,D_{KL}[\pi_{\eta_0} : \pi_{\mu_\eta^{(\lambda)}}] + (1 - \lambda)D_{KL}[\pi_{\eta_1} : \pi_{\mu_\eta^{(\lambda)}}] \tag{69}$$

$$= \lambda\,D_{\psi^*}[\eta_0 : \mu_\eta^{(\lambda)}] + (1 - \lambda)D_{\psi^*}[\eta_1 : \mu_\eta^{(\lambda)}] \tag{70}$$

$$= \lambda\,\psi^*(\eta_0) + (1 - \lambda)\,\psi^*(\eta_1) - \psi^*\big(\lambda\,\eta_0 + (1 - \lambda)\,\eta_1\big) \tag{71}$$

where $\mu_\eta^{(\lambda)} = \lambda\,\eta_0 + (1 - \lambda)\,\eta_1$. This matches the geometric Jensen-Shannon divergence of Nielsen [36], whereas $\mathcal{J}_{\psi,\alpha}$ in (58) is referred to as the dual version.

## E.1   Rényi Divergence as a Jensen Gap

We consider the Rényi $\alpha$ divergence between any two distributions $\pi_{\beta_1}$ and $\pi_{\beta_0}$ in our exponential family, so that $\pi_\beta(z|x) = \pi_0(z)^{1-\beta}\pi_1(z)^\beta / Z_\beta(x)$. Noting that the scaling factor $\alpha - 1 \leq 0$, we proceed to show that the scaled divergence is equal to a gap in Jensen's inequality:

$$(1 - \alpha)D_\alpha[\pi_{\beta_1}(z) : \pi_{\beta_0}(z)]$$

$$= (1 - \alpha)\frac{1}{\alpha - 1}\log\int \pi_{\beta_0}^{1-\alpha}\pi_{\beta_1}^\alpha dz$$

$$= -\log\int \Big(\frac{\pi_0^{1-\beta_0}\pi_1^{\beta_0}}{Z_{\beta_0}}\Big)^{1-\alpha}\Big(\frac{\pi_0^{1-\beta_1}\pi_1^{\beta_1}}{Z_{\beta_1}}\Big)^\alpha dz$$

$$= -\Big(\log\int \pi_0^{1-\beta_0-\alpha+\alpha\beta_0+\alpha-\alpha\beta_1}\pi_1^{\beta_0-\alpha\beta_0+\alpha\beta_1}dz - \big((1 - \alpha)\log Z_{\beta_0} + \alpha\log Z_{\beta_1}\big)\Big)$$

$$= -\Big(\log\int \pi_0^{1-[(1-\alpha)\beta_0+\alpha\beta_1]}\pi_1^{(1-\alpha)\beta_0+\alpha\beta_1}dz - \big((1 - \alpha)\log Z_{\beta_0} + \alpha\log Z_{\beta_1}\big)\Big)$$

$$= (1 - \alpha)\psi(\beta_0) + \alpha\psi(\beta_1) - \psi\big((1 - \alpha)\beta_0 + \alpha\beta_1\big)$$

$$= \mathcal{J}_{\alpha,\psi}$$

## E.2 Chernoff Point and Maximizing the Jensen Gap

In this section, we derive the optimal solution for the optimization defining the Chernoff information point. In particular, we optimize the Bregman Information or Jensen gap $\mathcal{J}$

$$\mathcal{J}_{\psi,\alpha} = \max_{\alpha} \min_{\beta_r} (1 - \alpha) \, D_\psi[\beta_0 : \beta_r] + \alpha \, D_\psi[\beta_1 : \beta_r] \tag{72}$$

$$= \max_{\alpha} (1 - \alpha) \, \psi(\beta_0) + \alpha \, \psi(\beta_1) - \psi((1 - \alpha)\beta_0 + \alpha\beta_1) \tag{73}$$

where we use arbitrary endpoints $\beta_0, \beta_1$ and mixing parameter $\alpha$ to highlight the arithmetic mean in the argument of the final term. This will match (38) when $\beta_0 = 0$, $\beta_1 = 1$ and $t = \beta$.

Now, we can differentiate with respect to $\alpha$, letting $\beta_\alpha = (1 - \alpha)\beta_0 + \alpha\beta_1$. We use the product rule and the identity $d\psi(\beta)/d\beta = \eta_\beta$ in the last term to obtain

$$\frac{d\mathcal{J}}{dt} = 0 = -\psi(\beta_0) + \psi(\beta_1) - \eta_{\beta_\alpha} \cdot (\beta_1 - \beta_0) \tag{74}$$

$$\implies \quad \eta_{\beta_\alpha} = \frac{\psi(\beta_1) - \psi(\beta_0)}{\beta_1 - \beta_0} \tag{75}$$

where $\eta_{\beta_\alpha}$ indicates the expected sufficient statistics, or dual parameter, corresponding to the natural parameter $\beta_\alpha = (1 - \alpha)\beta_0 + \alpha\beta_1$.

## E.3 Equal KL Divergences Derivation

We show that the KL divergences that constitute $\mathcal{J}_{\alpha,\psi}$ are equal at the critical point $\eta_\alpha = \frac{\psi(\beta_1) - \psi(\beta_0)}{\beta_1 - \beta_0}$:

$$D_\psi[\beta_0 : \beta_\alpha] = \psi(\beta_0) - \psi(\beta_\alpha) - (\beta_0 - \beta_\alpha)\eta_\alpha$$

$$= \psi(\beta_0) - \psi(\beta_\alpha) + \frac{(\beta_\alpha - \beta_0)}{\beta_1 - \beta_0}(\psi(\beta_1) - \psi(\beta_0))$$

$$= \frac{1}{\beta_1 - \beta_0}\left((\beta_1 - \beta_0)\psi(\beta_0) - (\beta_1 - \beta_0)\psi(\beta_\alpha) + (\beta_\alpha - \beta_0)\psi(\beta_1) - (\beta_\alpha - \beta_0)\psi(\beta_0)\right)$$

$$= \frac{1}{\beta_1 - \beta_0}\left((\beta_1 - \beta_\alpha)\psi(\beta_0) + (\beta_\alpha - \beta_0)\psi(\beta_1) - (\beta_1 - \beta_0)\psi(\beta_\alpha)\right)$$

$$= \left(\frac{\beta_1 - \beta_\alpha}{\beta_1 - \beta_0}\psi(\beta_0) + \frac{\beta_\alpha - \beta_0}{\beta_1 - \beta_0}\psi(\beta_1) - \psi(\beta_\alpha)\right)$$

$$D_\psi[\beta_1 : \beta_\alpha] = \psi(\beta_1) - \psi(\beta_\alpha) - (\beta_1 - \beta_\alpha)\eta_\alpha$$

$$= \psi(\beta_1) - \psi(\beta_\alpha) - \frac{(\beta_1 - \beta_\alpha)}{\beta_1 - \beta_0}(\psi(\beta_1) - \psi(\beta_0))$$

$$= \frac{1}{\beta_1 - \beta_0}\left((\beta_1 - \beta_0)\psi(\beta_1) - (\beta_1 - \beta_0)\psi(\beta_\alpha) - (\beta_1 - \beta_\alpha)\psi(\beta_1) + (\beta_1 - \beta_\alpha)\psi(\beta_0)\right)$$

$$= \frac{1}{\beta_1 - \beta_0}\left((\beta_1 - \beta_\alpha)\psi(\beta_0) + (\beta_\alpha - \beta_0)\psi(\beta_1) - (\beta_1 - \beta_0)\psi(\beta_\alpha)\right)$$

$$= \left(\frac{\beta_1 - \beta_\alpha}{\beta_1 - \beta_0}\psi(\beta_0) + \frac{\beta_\alpha - \beta_0}{\beta_1 - \beta_0}\psi(\beta_1) - \psi(\beta_\alpha)\right)$$

We have shown that the two divergences are equal when our condition on $\eta_\alpha$ holds. Further, observe that each divergence amounts to a Jensen gap $\mathcal{J}_{\alpha,\psi}$ with $\alpha = \frac{\beta_\alpha - \beta_0}{\beta_1 - \beta_0}$: This is more apparent for $\beta_0 = 0$ and $\beta_1 = 1$, where this simplifies using $\alpha = \frac{\beta_\alpha - \beta_0}{\beta_1 - \beta_0} = \beta_\alpha$:

$$D_\psi[\beta_0 : \beta_\alpha] = D_\psi[\beta_1 : \beta_\alpha]$$

$$= (1 - \beta_\alpha)\psi(0) + \beta_\alpha\psi(1) - \psi(\beta_\alpha)$$

$$= (1 - \beta_\alpha) \cdot 0 + \beta_\alpha \log p(x)$$
$$- \beta_\alpha \log p(x) + (1 - \beta_\alpha)D_{\beta_\alpha}[\pi_1(z|x) : \pi_0(z|x)]$$

$$= (1 - \beta_\alpha)D_{\beta_\alpha}[\pi_1(z|x) : \pi_0(z|x)],$$