# OpenReview forum: "Likelihood Ratio Exponential Families"
_NeurIPS.cc/2020/Workshop/DL-IG — NeurIPSW 2020: DL-IG Poster_

### Official Review · AnonReviewer1 · 2020-10-26
**Review of "Likelihood Ratio Exponential Families"**

**Rating:** 7
**Confidence:** 4

**Review:**

This paper considers distributions along a particular interpolating path between two distributions. This family of distributions can be interpreted as an exponential family defined by the likelihood ratio between the two distributions. The authors investigate the prevalence of distributions from this family in various problems of interest including rate-distortion with a logarithmic distortion, the information bottleneck, and a mixture of the two. The authors also investigate problems in hypothesis testing and connect the likelihood exponential family to the optimal error rates. I think this is a good paper, both in the background it provides and the new connections it makes.

---

### Decision · Program_Chairs · 2020-11-07

Accept (Poster)